# Psychosocial Factors Associated with Burnout and Self-Perceived Health in Spanish Occupational Therapists

**DOI:** 10.3390/ijerph20010044

**Published:** 2022-12-20

**Authors:** Rubén Juy, Ana Nieto, Israel Contador, Francisco Ramos, Bernardino Fernández-Calvo

**Affiliations:** 1Department of Personality, Assessment, and Psychological Treatments, Facultad de Psicología, University of Salamanca, 37005 Salamanca, Spain; 2Department of Basic Psychology, Psychobiology, and Methodology of Behavioral Sciences, Facultad de Psicología, University of Salamanca, 37005 Salamanca, Spain; 3Department of Psychology, University of Cordoba, 14071 Cordoba, Spain; 4Maimonides Biomedical Research Institute of Cordoba (IMIBIC), University of Cordoba, 14004 Cordoba, Spain; 5Department of Psychology, Federal University of Paraiba, Joao Pessoa 58051-900, Brazil

**Keywords:** burnout, occupational therapist, emotional exhaustion, health, stress, coping, personality

## Abstract

(1) Background: There are few studies of burnout syndrome (BS) in occupational therapists (OTs), and protective factors for BS have received little attention in the scientific literature. This research aimed to estimate the prevalence of BS, characterize the associated psychosocial factors, and analyze their relationship with health in a sample of Spanish OTs. (2) Methods: A total of 127 therapists completed the Maslach Burnout Inventory (MBI) and other standardized questionnaires measuring: personality traits (reduced five-factor personality inventory, NEO-FFI), coping styles (Coping Strategies Questionnaire, CAE), work-family conflict (Survey Work-Home Interaction Nijmegen, SWING), professional factors (role ambiguity/clarity and modified role conflict questionnaires), and the perception of health (Goldberg’s General Health Questionnaire). Several correlational and multiple regression analyses were performed to study the psychosocial predictors of burnout and its relationship with health perceptions. (3) Results: 15.8% of the professionals presented BS, with emotional exhaustion (EE; 38.7%) being the most compromised dimension. Neuroticism, role conflict, negative work-family interaction, and open emotional expression (OE) significantly predicted a higher EE. The main predictors of cynicism (CY) were being male, role conflict, and OE. Role conflict, role ambiguity and social support-seeking were significant predictors of reduced professional efficiency (PE). (4) Conclusions: A high percentage of OTs with BS suggests the need for increased awareness of the importance of this syndrome in the health community. It would be critical to consider the protective factors (i.e., emotional management, social support) that help promote OTs’ well-being and health.

## 1. Introduction

Occupational therapists (OTs) are healthcare workers who promote engagement in meaningful activities as a means of enhancing users’ well-being and health [1]. Health professions, oriented to the care of people’s health and well-being, are associated with high rates of burnout [2,3,4], an organism’s response to sustained work stress [5]. Various theoretical models [6,7,8] consider burnout syndrome (BS) as the result of a problematic or unbalanced relationship between a person’s resources and the work situation. Essentially, BS is characterized by the combination of three types of manifestations or dimensions: emotional exhaustion (EE; i.e., wear and tear, emotional and physical exhaustion, loss of energy, weakening and fatigue), cynicism (CY; i.e., negative or inappropriate attitudes, irritability and detached concern toward people and different work aspects), and inefficacy or reduced professional efficiency (RPE; i.e., negative self-evaluation of competence, achievements, and productivity at work) [9]. BS negatively affects personal health (physical and psychological) and the quality of care that health professionals provide [3,4]. 

In particular, various studies report high percentages of burnout among OTs, although the prevalence rates vary considerably. In Canada, about one-third of OTs have burnout symptoms; specifically 34.8%, 43.5%, and 27.5% show EE, CY, and RPE, respectively [10]. In some countries similar to our (Spanish) environment, 44.7% of Portuguese OTs have job burnout, as measured by the cut-off point (≥50) of the Copenhagen Burnout Inventory [11]. In Greece, Katsiana et al. [12] observed that 54% of OTs have moderately high or high scores in all three dimensions of the Maslach Burnout Inventory (MBI) (51% EE, 65% CY, and 48% RPE). In Spain, Escudero-Escudero et al. [13], using different criteria from those of the normative study of the MBI (39), found that 69.4% present at least some manifestation of burnout: EE (63.5%), CY (33.9%), and RPE (2.1%); only 1.8% of the OTs had all three dimensions compromised. In this context, the measuring instruments, the statistical methods applied, the sociocultural context, and the quality of the studies (e.g., sample size) may explain the heterogeneity of the figures [9,14]. 

Understanding the factors responsible for burnout is crucial for promoting health and well-being. The models differ in the weight given to the components (individual vs. situational) when explaining these factors, but agree that burnout is a dynamic and transactional process [15]. Thus, some studies show that burnout is significantly associated with sociodemographic variables, such as age [11,13,16], gender [16], or civil status [17]. However, others emphasize the role of situational and psychological factors [18]. Thus, different investigations highlight the role of the work environment in the development of burnout [13,19,20,21]: high workload [22], type of shift [13], the presence of role conflicts or role ambiguity [23,24,25]. In fact, some researchers [8,26] report six areas of work that may cause burnout: job workload (excessive tasks/work hours), control (role conflict or ambiguity), reward (monetary and intrinsic rewards), community (social interaction, support), fairness (job turnover), and values (moral disagreements, value incongruity). Finally, the work-family interaction is also a central factor in the development of burnout. For example, the negative balance between work and family is associated with lower psychological and physical well-being [27,28] and higher levels of burnout in several public collectives [28,29,30,31].

Unlike situational factors, psychological factors have awakened less empirical interest [13,18,32]. Thus, certain personality traits are associated with burnout level [33,34,35], and neuroticism is one of the best predictors of BS [34,36]. Moreover, the positive effects of specific psychological resources (e.g., resilience, grounded optimism) to overcome stress and build positive states have been tested in different populations [37,38], but investigations related to personal resources (e.g., stress coping, social support) that could buffer stress and its consequences in OTs are very scarce. Coping strategies comprise continuous efforts (emotional, cognitive, and behavioral) to cope with the environment’s demands [15]. In different health professions, it has been observed that problem-focused coping correlates negatively with the three burnout dimensions, whereas emotion-focused coping correlates positively [34,39,40]. Gupta et al. [10] found that OTs with high professional efficiency tended to use strategies based on social support (e.g., “keeping in touch with referral networks”).

The objective of this study was to estimate the prevalence of burnout in a sample of Spanish OTs and analyze the factors (sociodemographic, psychological, social, and work-related) associated with its manifestations (EE, CY, and RPE). The relationship between the dimensions of burnout and the perception of health was also analyzed, controlling for the effect of several covariates in the model. 

## 2. Materials and Methods

### 2.1. Participants 

A convenience sample of 127 OTs from different geographical areas of Spain was analyzed. All participants were affiliated to three regional official colleges of OTs in the North of Spain (Aragón, Cataluña, and Castilla y León), which agreed to take part in the study. The selected individuals were volunteers who signed a written informed consent before their enrollment in the study. The study was approved by the institutional review board of each OT college, and conducted following the standard guidelines for human research established in the Declaration of Helsinki [41].

### 2.2. Instruments

Participants completed a questionnaire that collected sociodemographic data (age, sex, civil status), aspects related to professional context (i.e., work environment, work sector), and several standardized questionnaires evaluating psychological, family, professional, and health factors. The main characteristics of the applied instruments are described in this section.

Maslach Burnout Inventory (MBI) [42]. The MBI includes 22 items that evaluate feelings and attitudes toward work according to the three subscales/dimensions of the questionnaire. Each item is scored on a seven-point Likert-type scale, ranging from 0 (never) to 6 (every day). High scores in EE and CY dimensions and low scores in professional efficiency (PE) correspond to BS (38). This questionnaire was validated in the Spanish population by Gil-Monte and Peiró [43] and the subscales’ reliability (Cronbach’s alpha [α]) ranged from 0.72 to 0.87.

Five-factor Reduced Personality Inventory (NEO-FFI) [44]. This instrument contains 60 items distributed in 5 personality factors: Neuroticism (N), Extraversion (E), Openness to Experience (O), Agreeableness (A), and Conscientiousness or Responsibility (C). Items are rated on a five-point Likert-type scale ranging from 1 (strongly disagree) to 5 (strongly agree). The dimensions have a maximum score of 60, with higher scores indicating greater factor intensity. The Spanish version was validated by Manga et al. [45] and the reliability values (α) varied between 0.71 and 0.82. 

Work-Home Interaction-Nijmegen Questionnaire (SWING) [46]. The Spanish version of the SWING was validated by Moreno-Jiménez et al. [47]. This instrument evaluates the reconciliation of home-work life with 22 items scored on a Likert format from 0 (never) to 3 (always). The SWING presents 4 subscales evaluating positive and negative work-home relationships. The total score of the subscales ranges from 0 to 24, with higher scores indicating a greater intensity (positive or negative) of the interactions. The reliability (α) of the scales oscillated from 0.77 to 0.89. 

The Role Ambiguity General Questionnaire [48]. The Spanish version, validated by Peiró et al. [49], presents 30 items measuring role ambiguity/clarity (15 even items) and role conflict (15 odd items). It is rated on a seven-point Likert-type scale ranging from 1 (strongly disagree) to 7 (strongly agree) [22]. Subscale scores range from 1 to 105, with higher scores indicating greater role clarity and role conflict, respectively. The reliability values (α) of the original scale were 0.87 and 0.82 for role ambiguity and role conflict, respectively.

Coping Strategies Questionnaire (CAE) [50]. The CAE is an original test developed in Spain that evaluates coping style through 42 items, divided into 7 subscales: Problem-solving focused coping (PS), Negative self-focus (NF), Positive reappraisal (PR), Open emotional expression (OE), Avoidance (A), Social support-seeking (SS) and Religion (RL). Items are scored on a five-point Likert-type scale, ranging from 0 (never) to 4 (almost always). High scores indicate a higher intensity of the coping style employed. The alpha values of the subscales ranged from 0.64 to 0.92.

Goldberg General Health Questionnaire (GHQ-12), reduced 12-item version [51]. The GHQ-12 is a self-report instrument developed to detect psychological maladjustment and psychiatric disorders in community settings (12). The Spanish version of the GHQ-12 by Rocha and Rodríguez-Sanz [52] consists of 12 items rated on a Likert format (0–3). The total score ranges from 0 to 36, with higher scores indicating poorer health. The questionnaire showed a high reliability (internal consistency) in the general population (α = 0.86).

### 2.3. Design and Procedure

This is an ex post facto, descriptive-correlational single-group study. To complete the data collection, the questionnaires were adapted for a Google Drive form. After this, they were disseminated through an URL, using different social and media technological media (email, college website, Facebook…), among three Spanish OT colleges that agreed to participate in the study. The questionnaires were answered in single attempt and the time of response was thirty minutes approximately. After the six months established for the receipt of the responses, the data collection was closed to start with the processing. 

### 2.4. Statistical Analysis

First, a descriptive analysis of the sample (means, standard deviations, and percentages) was performed, and subsequently, the prevalence of BS in the sample was estimated. The reference values adopted to establish disturbance were ≥25 for EE, ≥9 for CY, and ≤35 points for RPE [53]. Moreover, the cut-off points of the 66% of the participants above (EE and CY dimensions) and the 33% of the participants below (PE dimension) the distribution were also calculated. Thus, values ≥27 and ≥9 points in EE and CY, respectively, were considered high, whereas a score of ≤35 in PE was considered low. Next, correlational analyses, with the Pearson (r) or point-biserial (r_bp_) correlation coefficients, were used to analyze the relationships between burnout dimensions and the sociodemographic, psychological, family, professional, and health variables included in the study. Subsequently, several multiple regression analyses (ENTER method) were performed to predict each dimension of burnout (outcomes). The variables significantly related to each burnout dimension were selected as potential predictors (*r* > 0.20). Finally, the burnout dimensions predicting health status were selected, controlling for the effect of different covariates related to OTs health (*r* > 0.20). In all the multiple regression analyses, the Durbin–Watson (D-W) statistic and the variance inflation factor (VIF) were calculated to analyze problems of residual independence and multicollinearity between the variables. Values of DW between 1.5 and 2.5 and of VIF ≤ 5 were established to assume residual independence [54] and the variables’ absence of collinearity [55], respectively.

The Statistical Package for Social Sciences (SPSS) version 25.0 (IBM Corp. Chicago, US, Released 2013) was used for the statistical analyses. All estimates and inferences from statistical tests were considered significant when the *p*-value was <0.05.

## 3. Results

Table 1 shows the sociodemographic and professional characteristics of the sample This study comprised 127 OTs from 21 to 62 years (mean [M] = 33.66, standard deviation [SD] = 8.66 years). Most of them were single young women with no children, assuming household responsibilities, and usually did not support dependent people economically. In general, they were dedicated exclusively to their profession; many worked in a single work-place as professional of reference in the field (i.e., without other OT colleagues in the team). The private work sector was the most frequent, with geriatrics being the most frequent work area. Concerning professional experience, mostly OTs had been more than 5 years working with full permanent contracts, but more than half believed their pay was low or very low.

### 3.1. Prevalence of Burnout

The professionals obtained the following average scores on the MBI: EE (*M* = 21.69, *SD* = 12.37), CY (*M* = 7.26, *SD* = 5.32), and PE (*M* = 37.98, *SD* 5.51). Of the sample of OTs, 15.7% presented BS according to all three dimensions (high EE, high CY, and RPE), whereas 17.3% presented BS according to two dimensions (EE and CY in 6.5%). Further, 24.4% of the sample presented symptoms in a single dimension (38.7%, 35.5%, and 25.8% in EE, CY, and RPE, respectively). Lastly, 42.5% of the OTs manifested no problems in any BS dimension. The prevalence of BS (high EE, high CY, and RPE) was similar (15.7%) when using the cut-off points calculated in the selected sample. 15% of the cases presented two affected dimensions, and in 5.5% of these cases, the EE and CY dimensions were implicated. In summary, around 15% of OTs showed serious work stress (i.e., three burnout dimensions were affected), and near of 25% had, at least, one troubled dimension. 

### 3.2. Relationship between Psychosocial Factors and Burnout Dimensions

Table 2 shows the descriptive analyses (psychosocial, family, and professional factors) and their correlations with the burnout dimensions. As shown, OTs primarily used coping strategies based on adaptive behaviors: SS, PS, and PR. When examining family-work interactions, we observed that the sum of positive home-work interactions (PHWI) obtained the highest score, followed by negative work-home interactions (NWHI). Concerning personality traits, we observed that neuroticism (N) obtained the lowest mean score (*M* = 34.24, *SD* = 5.17), unlike conscientiousness (C) (*M* = 42.43, *SD* = 3.83), which obtained the highest score.

When analyzing the relationship between the MBI dimensions, we observed a moderate positive correlation between EE and CY (*r* = 0.557, *p* < 0.001), whereas PE showed moderate negative correlations with EE (*r* = −0.457, *p* < 0.001) and CY (*r* = −0.518, *p* < 0.001). The number of working hours (r = 0.187, *p* < 0.035) and salary dissatisfaction (r = −0.208, *p* < 0.019) were weakly related to EE, while working in the geriatric area (*r* = 0.203, *p* < 0.022) and being male (*r* = −0.201, *p* = 0.024) were weakly associated with CY. Regarding coping techniques, OE and NF presented weak/moderate significant correlations with the burnout dimensions, positive with EE and CY, and negative with PE. The SWING questionnaire yielded positive correlations for EE and CY (moderate and weak, respectively) when the home-work interaction was negative, whereas positive work-home interactions had a weak negative correlation with EE. In addition, when the home-work interaction was positive, higher PE scores were observed (weak association). Concerning role stress, role clarity was positively correlated with PE and negatively with EE, whereas the correlations between role conflict and the three burnout dimensions were positive (stronger for EE and CY). All these correlations were from weak to moderate. Finally, neuroticism was positively related to higher EE and CY (weak and moderate associations), unlike PE, whose correlation with neuroticism was weak and negative. In brief, neuroticism (personality trait), NF and OE (coping strategies), negative interactions (work-home and home-work) and role conflict were consistently associated with the three burnout dimensions, being positively associated with EE and CY, and negatively to PE.

### 3.3. Multiple Regression Analysis

Table 3 presents the three multiple regression models to predict burnout dimensions. The EE dimension of burnout was predicted by working hours, role conflict, negative work-home interaction, neuroticism, and OE coping, which explain 58.5% of the total variance. The NI work-home (Beta = 0.39), neuroticism (Beta = 0.23) and role conflict (B = 0.21) were the best predictors. Similarly, being male, role conflict, and mainly OE coping (Beta = 0.28) were significant predictors of CY, explaining 34.5% of the total variance. Finally, the presence of role conflict and role ambiguity, as main predictors, was negatively related to PE, whereas a SS coping strategy was positively associated with PE. These factors explained 39.7% of the total variance associated with PE. In summary, the specific predictors showed a higher explanatory weight in the EE model versus CY and PE models. Basically, these models show that burnout, especially the EE dimension, is a product of sociodemographic, psychological, social, and work-related factors. The values of the D-W and VIF statistics indicated the independence of the residuals and the absence of multicollinearity, respectively.

### 3.4. Relationship of Burnout and General Health 

All the dimensions of burnout significantly correlated with a poorer health status: EE (*r* = 0.563, *p* < 0.001), CY (*r* = 0.283, *p* < 0.001), and PE (*r* = −0.296, *p* < 0.001). Likewise, the health status of the OTs was related to different psychosocial factors: role ambiguity (*r* = 0.278, *p* < 0.001), negative home-work interaction (*r* = 0.358, *p* < 0.001), neuroticism (*r* = 0.460, *p* < 0.001), FN (*r* = 0.398, *p* < 0.001), and OE coping (*r* = 0.270, *p* < 0.001). As is shown in Table 4, the multiple regression indicated that the EE dimension, a negative home-work interaction, and neuroticism contributed statistically and significantly to the prediction of health perception when controlling for the effect of sociodemographic factors (age and sex) and the other burnout dimensions (CY and PE). Thus, the OTs health is mainly affected by the EE dimension and other specific factors that are also associated with this burnout dimension. The VIF and D-W values of the regression models were satisfactory. 

These findings describe the rates of BS in Spanish OTs, as health professional workers. Moreover, we addressed the nature of main predictors of BS in OTs and the role of each burnout dimension on their health perceptions.

## 4. Discussion

The present research estimated the prevalence of burnout in a sample of Spanish OTs, as well as the main factors associated with BS. The results indicated that 15.7% of the OTs presented burnout with all three dimensions affected conjointly; 17.3% showed burnout on two scales (6.5% EE and CY, namely, the core of burnout); and 24.4% presented a single affected dimension (38.7%, 35.5%, and 25.8% for EE, CY, and RPE, respectively). Other investigations carried out in Spain with OTs indicate that 1.8% of OTs present burnout, whereas 26.1% present EE and CY [13]. The variability in the estimates of burnout prevalence seems to be related to the measurement instruments, the method of estimating burnout, and the sociodemographic aspects of the sample [14]. Escudero-Escudero et al. [13] used a different burnout estimation method, based on a sample of social workers, making comparisons difficult. In contrast, the present study used the cut-off points established in the normative criteria of the MBI in Spain [43].

Concerning the different psychosocial predictors related to burnout, we found that role conflict, negative work-home interaction, neuroticism, and OE-based coping predicted EE. Consistently, CY was associated with role conflict and OE-based coping, and with being male. Finally, role conflict and role ambiguity were negatively related to RPE, whereas a SS coping strategy was positively associated with PE. Accordingly, model 1 for EE explained 58.5% of the variance, while reported variance for model 2 (CY) and 3 (PE) was 34.5% and 39.7%, respectively. It is observed that the shared variables in these models (Neuroticism, OE, NF, Role Conflict and NI) explained the EE better than CY and PE dimensions. Previous works have reported that overwork [22,32], role conflict and role ambiguity [23,24,25], negative work-home interaction [28,31], neuroticism [33,34,35], and the use of emotion-focused coping strategies [39,40] are associated with the onset of burnout. These same constructs are also relevant in determining burnout in our research sample of OTs. Similarly, other studies claim that social support positively affects coping with stress and preventing its negative consequences [57,58]. The only dimension associated with gender was depersonalization or CY, which is consistent with results indicating a higher prevalence of this dimension in men [29,59].

Of all the predictors, role conflict was the only common predictor of all three dimensions, reinforcing its impact on burnout in OTs [24]. In contrast, role clarity only predicted a better perception of PE, with a less decisive role in burnout, as other studies have indicated [30]. Negative work-home interaction also affected EE negatively, as previously demonstrated [29]. Concerning psychological variables, only neuroticism or lack of emotional stability was associated with EE, but not the other personality factors [33]. This study also confirm that emotion-focused strategies are not the best option for coping with stress [39,40], whereas social support contributes to greater professional effectiveness [10].

Finally, the results indicated that EE was the only burnout dimension that predicted the general health of OTs, apart from negative home-work interaction and neuroticism, after considering the sociodemographic variables and the other burnout dimensions in the model. Previously, other studies have shown that burnout is associated with workers’ poorer physical and mental health [60]. In fact, as we have seen, negative work-home interactions and neuroticism are associated with greater EE, the main dimension related to the general health of the OTs we evaluated. These results are consistent with other studies pointing out that EE is significantly and positively associated with the loss of subjective well-being [61].

This research has theoretical and practical implications. Theoretically, it proves that burnout is a multifactorial syndrome, that involves socio-demographics (i.e., gender in CY), psychological characteristics (neuroticism and ineffective coping strategies), family relationships, and aspects linked to the professional role (lack of clarity and role conflict). On a practical level, these data (i.e., 15% of OTs suffering burnout) alert us to the need to promote health and well-being in this health professional collective. At individual level, initiatives related to the management of emotions and stress programs, as well as the implementation of appropriate coping strategies, are recommended. At organizational level, it is important to promote work-family conciliation and clear structures of task responsibilities to avoid the generation of conflicts or role misunderstandings.

This study has a series of limitations that should be commented upon. Firstly, the convenience sampling methods and the reduced sample size limits the generalizability to the whole OTs population in Spain. In addition, the sample was composed mainly of women, while men were underrepresented. We note that women are very numerous in the group of OTs, as indicated by the study with the largest sample of OTs in our country [13]. Secondly, the study is based on a self-report methodology, although all the questionnaires used have been validated in the Spanish population and have adequate psychometric properties. Finally, the results were obtained with a cross-sectional design and using a limited number of variables because diverse factors are associated with burnout at the psychological, family, and professional levels. Future research should look into other psychosocial predictors of burnout in OTs, introducing protective psychological resources (e.g., optimism, resilience, etc.). We also recommend a prospective methodology, which allows analyzing the natural course or development of burnout. The design and implementation of more OT-specific assessment instruments (i.e., quality of praxis or specific stressor delimitations) is also an aspect to consider in future studies.

## 5. Conclusions

In terms of prevalence rates, 15% of the OTs showed a full burnout syndrome, while 25% of them had at least one burnout dimension affected. These findings alert us to the importance of BS in OTs and provide new evidence to better understand its development. Thus, different psychosocial factors (neuroticism, ineffective coping strategies, negative work-family interaction, etc.), beyond the role of objective stressors (e.g., working hours), are implicated on the BS in the OTs’ collective. Finally, organizations and workplaces need to become aware of the impact of BS on professionals’ health, which may have implications on the provided quality of care and organizations’ effectiveness. The development of intervention actions or programs, that prevent BS occurrence, is a cornerstone in caring for both the professionals’ and our own health.

## Figures and Tables

**Table 1 ijerph-20-00044-t001:** Sociodemographic and professional characteristics of the sample (N = 127).

Variables	*n*	Percentage
Sex		
Women	118	92.9
Men	9	7.1
Marital status		
Single	85	66.9
Married	37	29.1
Divorced	4	3.1
Others	1	0.8
Children (*n*)		
0	86	66.7
1	15	11.8
2	23	18.1
3	3	2.4
Household chores		
Somewhat	10	7.9
Quite a lot	19	15.0
Almost always	22	17.3
Always	76	59.8
Care of dependent people (*n*)		
0	81	63.8
1	18	14.2
2	23	18.1
3 or more	5	3.9
Work dedication to OT		
Exclusive, No	18	14.2
Exclusive, Yes	109	85.8
Work centers (*n*)		
1	84	66.1
2	33	26.0
3 or more	10	7.9
Work locations (*n*)		
1	101	79.5
2	18	14.2
3 or more	8	6.3
Work sector		
Public	33	26.0
Private	87	68.5
Self-employed	7	5.5
Work experience		
Less than 1 year	8	6.3
Between 1 and 3 years	27	21.3
Between 3 and 5 years	10	7.9
More than 5 years	82	64.6
Type of contract		
Internship	3	2.4
Temporary	37	29.1
Indefinite	87	68.5
Weekly working hours		
From 3 to 5 h	1	0.8
5 to 10 h	4	3.1
10 to 20 h	9	7.1
20 to 30 h	18	14.2
30 to 40 h	95	74.8
OTs at the work center (*n*)		
0	81	63.8
1	19	15.0
2	10	7.9
3 or more	17	13.4
Working Field		
Pediatrics	13	10.2
Geriatrics	64	50.4
Mental disability	9	7.1
Physical disability	18	14.2
Others	23	18.1
Salary perception		
Very low	25	19.7
Low	46	36.2
Normal	46	36.2
High	10	7.9

*n* = number of individuals; OT = Occupational Therapy; OTs = Occupational Therapists.

**Table 2 ijerph-20-00044-t002:** Mean, standard deviations, and correlations between psychological, family, and work factors with the dimensions of the Maslach Burnout Inventory (MBI; *n* = 127).

Variables	*M (SD)*	EE	CY	PE
NEO-FFI				
Neuroticism	34.24 (5.17)	0.46 **	0.30 **	−0.29 **
Extraversion	37.52 (2.96)			
Openness	37.44 (4.79)			
Agreeableness	35.31 (4.23)			
Conscientiousness	42.43 (3.83)	0.25 **		
CAE				
PSF	16.80 (3.76)			0.20 *
NF	8.32 (3.85)	0.39 **	0.33 **	−0.37 **
PR	15.95 (3.35)			
OE	8.35 (3.31)	0.33 **	0.37 **	−0.24 **
SS	15.76 (5.88)			0.21 *
Religion	1.59 (3.95			
Avoidance	12.26 (4.01)			
SWING	8.65 (5.31)			
NI (Work-Home)	2.04 (1.94)	0.60 **	0.25 **	−0.24 **
NI (Home-Work)	8.39 (3.56)	0.19 *	0.22 *	−0.20 *
PI (Work-Home)	9.97 (3.57)	−0.23 **		0.17 *
PI (Home-Work)	8.65 (5.31)	−0.23 *		0.28 **
ROLE STRESS				
Role ambiguity/clarity	67.06 (7.65)	−0.23 *		0.46 **
Role conflict	62.18 (10.62)	0.41 **	0.36 **	−0.25 **

Note. EE = Emotional Exhaustion; CY = Cynicism; PE = Professional Efficiency; NEO-FFI = Personality questionnaire; CAE = Stress coping questionnaire; PSF = Problem-solving focused; NF = Negative self-focus; PR = Positive reappraisal; OE= Open emotional expression; SS = Social-seeking support; SWING = Survey Work-Home Interaction-Nijmegen; NI = Negative interactions; PI = Positive interactions. Significance values * *p* < 0.05. ** *p* < 0.01. For interpreting the correlations, the following ranges were followed [56]: weak (0.10–0.39), moderate (0.40–0.69), and strong (0.70–0.99).

**Table 3 ijerph-20-00044-t003:** Multiple regression models of burnout dimensions.

	*B*	Error	Beta	*t*	*p*	L Limit	U Limit	VIF
EE								
(Constant)	−19.80	9.25		−2.14	0.034	−38.10	−1.49	
Working hours ^ͳ^	4.59	1.76	0.16	2.61	0.010 *	1.10	8.08	1.10
Salary	−1.10	1.56	−0.04	−.71	0.481	−4.18	1.98	1.12
Role ambiguity	−0.17	0.10	−0.10	−1.68	0.095	−0.38	0.03	1.15
Role conflict	0.23	0.08	0.21	3.22	0.002 **	0.10	0.40	1.12
NI (Work-Home)	0.91	0.15	0.39	5.92	0.001 **	0.60	1.21	1.23
Neuroticism	0.51	0.17	0.23	3.26	0.001 **	0.21	0.87	1.35
OE	0.72	0.23	0.19	3.09	0.002 **	0.26	1.19	1.11
NF	0.29	0.23	0.09	1.30	0.196	−0.153	0.74	1.39
*F*_(8, 118)_ = 19.23. *p* < 0.001; *R^2^* = 0.585, *R*^2^_Adjusted_ = 0.557; D-W = 2.25								
CY								
(Constant)	−2.88	4.469		−0.64	0.521	−11.73	5.97	
Gender	−4.06	1.54	−0.20	−2.63	0.010 *	−7.12	−1.01	1.02
Work area ^#^	0.68	0.82	0.06	0.83	0.408	−0.94	2.29	1.07
Role conflict	0.13	0.04	0.26	3.34	0.001 **	0.05	0.21	1.13
NI (Work-Home)	0.06	0.08	0.06	0.80	0.428	−0.10	0.22	1.16
Neuroticism	0.10	0.09	0.10	1.12	0.267	−0.08	0.27	1.32
OE	0.46	0.13	0.28	3.62	0.001 **	0.21	0.70	1.11
NF	0.21	0.12	0.15	1.74	0.084	−0.03	0.44	1.34
*F _(_*_6, 120)_ = 8.93. *p* < 0.001; *R^2^* = 0.345, *R^2^*_Adjusted_ = 0.306; D-W = 2.17								
PE								
(Constant)	29.89	4.95		6.04	0.001	20.09	39.69	
Role ambiguity	0.30	0.06	0.41	5.34	0.001 **	0.19	0.41	1.16
Role conflict	−0.11	0.04	−0.22	−2.80	0.006 **	−0.19	−0.03	1.18
NI (Work-Home)	−0.00	0.08	−0.01	−0.03	0.980	−0.16	0.16	1.20
Neuroticism	−0.10	0.09	−0.01	−1.20	0.235	−0.28	0.07	1.33
OE	−0.25	0.13	−0.15	−1.92	0.057	−0.51	0.01	1.19
NF	−0.20	0.12	−0.14	−1.67	0.092	−0.44	0.03	1.39
SS	0.16	0.07	0.17	2.23	0.028 *	0.02	0.28	1.12
*F _(_*_7, 119)_ = 11.17. *p* < 0.001; *R^2^* = 0.397, *R^2^*_Adjusted_ = 0.361; D-W = 1.91								

Note. B = non standardized coefficient; Beta = Standardized coefficient; L = Lower; U = Upper; t = Student-t; VIF = Variance inflation factor; EE = Emotional Exhaustion; CY = Cynicism; PE= Professional Efficiency; ^ͳ^ Weekly hours (40 h vs. less than 40 h); ^#^ = Geriatrics Area vs. other areas; NI = Negative interaction; OE = Open emotional expression; NF = Negative self-focus; SS= Social-seeking support; *R^2^*= proportion of variance explained by predictor variables; *R^2^*_Adjusted =_ adjusted proportion of variance explained by predictor variables; D–W = Durbin-Watson statistic. Significance values * *p* < 0.05. ** *p* < 0.01.

**Table 4 ijerph-20-00044-t004:** Multiple regression for the dependent variable: General Health Questionnaire (GHQ-12).

	*B*	Error	Beta	*t*	*p*	L Limit	U Limit	VIF
(Constant)	13.40	7.21		1.86	0.066	−0.88	27.69	
Age	−0.03	0.05	−0.04	−0.67	0.504	−0.14	0.07	1.034
Gender	0.12	1.75	0.01	0.07	0.944	−3.34	3.59	1.08
EE	0.24	0.05	0.44	5.12	0.001 **	0.15	0.34	1.83
CY	−0.01	0.11	−0.01	−0.11	0.912	−0.24	0.21	1.91
PE	0.11	0.11	0.09	1.04	0.300	−0.10	0.32	1.83
Role ambiguity	−0.13	0.07	−0.14	−1.91	0.058	−0.26	0.01	1.35
NI (Family-Work)	1.083	0.25	0.31	4.43	0.001 **	0.60	1.57	1.19
Neuroticism	0.29	0.10	0.22	2.84	0.005 **	0.0	0.49	1.43
NF	0.12	0.14	0.07	0.87	0.388	−0.15	0.39	1.47
EA	−0.09	0.15	−0.04	−0.57	0.572	−0.38	0.22	1.34
*F _(_*_10, 116)_ = 18.85. *p* < 0.001; *R^2^* = 0.526, *R^2^*_Adjusted_ = 0.485; D-W = 1.98								

Note. GHQ-12 = Goldberg General Health Questionnaire; Beta = Standardized coefficient; t = Student-t; L = Lower; U = Upper; VIF = Variance inflation factor; EE = Emotional Exhaustion; CY = Cynicism; PE = Professional Efficiency; NI = Negative interaction; NF = Negative self-focus; OE = Open emotional expression; *R^2^*= proportion of variance; *R^2^*_Adjusted_ = Proportion of adjusted variance; D-W = Durbin–Watson statistic. Significance values. ** *p* < 0.01.

## Data Availability

The analyzed data is available under reasonable request.

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
