# Peer review of "Psychosocial Factors Associated with Burnout and Self-Perceived Health in Spanish Occupational Therapists"

_ijerph, 2022, doi:10.3390/ijerph20010044_

Round 1

Reviewer 1 Report

Firstly, I would like to add that this is a very accurate manuscript, which respects all the essential steps in performing a clear research. I would make some suggestions to improve it, some of them are my own personal preferences and some of them are mandatory.

Please, find below the suggestions of improving the manuscript:

-        The manuscript has a sound structure in which the author(s) has included in the Introduction section the information from the scientific literature regarding the burnout syndrome found in the occupational therapists (OTs) across the world such as Canada, Portugal, Greece, and Spain. So, the author(s) has provided information about the background of the case study. Going further with the introduction structure, the author(s) has identified correctly the gap in the literature by augmenting that the investigations related to personal resources ….to overcome for consequences in OTs are very scarce. As I have checked, the author(s) are right and it made me realize the importance of this piece of research. In addition, the author(s) has identified a case study by Escudero-Escudero et al. in which the Maslach Burnout Inventory has been applied in the Spanish population.

-        the Material and Methods section of the manuscript consists of subsections which would not address any further questions from me, but they may be improved. For instance, in the Participants subsection the author(s) specified that all respondents signed a written informed consent and that the study was approved by the participating professional associations. It would be appropriate to ask the author(s) for an Ethical approval number. Since this is the journal’s policy, I would ask the author(s) for the Ethical approval number to include it either in the Participants subsection or in the end of the manuscript in the Institutional Review Board Statement.

It would have also been useful for the author(s) to include the sampling method they have employed. If it was the convenience sampling method then in the limitations paragraph some limits of the sampling method should be added. But again, this is just my preference to include some information about the sampling method. I concluded it was the convenience sampling method after reading in the manuscript lines 101-102, respectively After contracting various professional territorial associations to disseminate the study. Another interesting fact which should be added related to the number of the initial OTs approached and what percentage represents the 127 OTs remained (the response rate).

The instruments used in collecting data were also translated in Spanish, and I guess they were all standardized for this population. Regarding the Coping strategies Questionnaire, the author(s) did not specify for it as being the Spanish version but I have checked the reference 49 and it was a reference in Spanish, namely, Sandin, B., Chorot, P (2003). Cuestionario de Afrontamiento Del Estres (CAE). In addition, the author(s) included the Cronbach’s alpha coefficients for each instrument, and since they were all in the correct range, there is no need to psychometrically validate them again, especially if the aim of the case study is not this one.

From the Design and procedure subsection, I have concluded that the instruments have been administered in several phases. I thought there was no need to elaborate more on the matter but maybe explaining what are the series of phases in more detail would be appropriate.

There is no further need to include other information in the statistical analyses subsection.

-                              The Results section is well-structured on subsection being supported by clear tables (Table 1, 2 and 3). All statistical significant correlations have been identified in the shape of correlated negatively with. My personal preference would have been to also include which type of Pearson correlations have been established, namely, high, moderate or low. But again, this is just my personal preference and I would not include it as long as the author(s) has correctly identified them.

-                              Regarding the Multiple regression analysis subsection, the author(s) has explained that he has used the Enter method and he has included all the necessary information for the reviewers to conclude whether their employment was appropriate or not. As such, the author(s) has included the Durbin-Watson test and the VIF statistics, which were all in range and the author(s) has also identified the statistical significant variables. Again, I highly suggest the author(s) to make some explanations for the R square values. For instance, the variables introduced in the multiple regression for the EE dependent variable explained 58,5% of the variance of EE (R square=0,585). For CY and PE, the explained variance of the variables is even less, namely, 34,5% and 39,7%. Thus, maybe it would be useful that some explanations regarding the R square values to be integrated in the Discussion section.

-                              - The Discussion section is strongly associated with the existing body of scientific literature. The author(s) has integrated in this section the limitations of the case study and not in the Conclusions section, which is very appropriate. However, by having a closer look at the first paragraph from the Discussion section, I have seen that some references are in () and some are in []. The author(s) should correct them accordingly but lines 270-272 should make reference to a Spanish piece of literature and reference 11 is about Portugal. Moreover, by looking at line 274, namely Escudero-Escudero et al. with the reference 11, I have realized that there is a mistake between the references as Escudero-Escudero is reference 13 and not reference 11. Thus, I highly recommend the author(s) to double check their references list and the numbers used in the body of the manuscript.

All in all, the points where the manuscript should be improved are the following:

  1. mandatory changes- check the references list, especially for the first paragraph in the Discussion section, add some explanations about the R square from the multiple regressions, and add the no. of the ethical approval
  2. recommended changes: add the sampling method plus the disadvantages in using it, the number of participants who were initially approached (response rate)

optional changes: add the types of Pearson correlations, namely high (more than 0.70), moderate (0,40-0,70) and low (below 0,40).

Reviewer 2 Report

Dear authors, thank you for the opportunity to read your comprehensive study.

The introduction describes in detail the relevance and scientific knowledge of the issue, and also determines the purpose of this study.

Materials and methods

Sample. 1. The description of the sample should include the age of the subjects and the length of service (minimum and maximum limits, average age and length of service), list how many medical institutions were covered by the study (in other words, be more specific).

2. Clarify whether the compliance of the study with the ethical principles of the Declaration of Helsinki was fixed in any document (at the moment this information is not specific).

The research instruments, procedure and statistical methods are described fully and specifically.

results

3. I would like to see in the table the distribution of surveyed employees regarding socio-demographic characteristics (not just a description of the prevailing groups).

4. It is necessary to add the interpretation of the results to tables 1, 2 and 3. Not only to list what statistically significant relationships were found, but also to reveal how they manifest themselves and are revealed (slightly expand the interpretation).

5. I would also like to see a summary of the relative research questions that the authors tested in their study at the end of the Results section.

The discussion of the results analyzes the findings in relation to previous studies, contains research limitations and future research directions.

6. Could the authors add - formulate the practical significance of their research, indicate how the results can be applied in practice, which authors can give practical recommendations.

7. The conclusion needs to be slightly specified in relation to the purpose of the study. At the moment it is not entirely based on the results obtained.

With respect for your work and best wishes, the reviewer
